# Identifying Whitemouth Croaker (*Micropogonias furnieri*) Populations along the Rio de Janeiro Coast, Brazil, through Microsatellite and Otolith Analyses

**DOI:** 10.3390/biology12030360

**Published:** 2023-02-24

**Authors:** Taynara Pontes Franco, Anderson Vilasboa, Francisco Gerson Araújo, Joana de Moura Gama, Alberto Teodorico Correia

**Affiliations:** 1Laboratório de Ecologia de Peixes, Departamento de Biologia Animal, Universidade Federal Rural do Rio de Janeiro (UFRRJ), Seropédica, Rio de Janeiro 23897-030, Brazil; 2Centro Interdisciplinar de Investigação Marinha e Ambiental (CIIMAR/CIMAR), Terminal de Cruzeiros do Porto de Leixões, Avenida General Norton de Matos S/N, 4450-208 Matosinhos, Portugal; 3Laboratório de Genética Pesqueira e da Conservação, Departamento de Genética, Instituto de Biologia Roberto Alcantara Gomes, Universidade do Estado do Rio de Janeiro (UERJ), Rua São Francisco Xavier 524, Maracanã, Rio de Janeiro 20550-900, Brazil; 4Departamento de Biologia Estrutural e Funcional, Universidade Estadual de Campinas (UNICAMP), Cidade Universitária, Campinas 13083-863, Brazil; 5Departamento de Biologia e Ambiente (DeBA), Escola de Ciências da Vida e do Ambiente (ECVA), Universidade de Trás-os-Montes e Alto Douro (UTAD), Quinta de Prados, 5001-801 Vila Real, Portugal; 6Departamento de Produção Aquática (DPA), Instituto de Ciências Biomédicas Abel Salazar (ICBAS), Universidade do Porto (UP), Rua de Jorge Viterbo Ferreira 228, 4050-313 Porto, Portugal

**Keywords:** Sciaenidae, population structure, molecular tags, geochemical and morphological signatures

## Abstract

**Simple Summary:**

The whitemouth croaker *Micropogonias furnieri* is an important fishery resource on the southwest Atlantic coast. Despite being heavily exploited, there are a few uncertainties regarding the population structure in the transition area between the tropical and warm temperate zones (South Brazilian Bight). In the State of Rio de Janeiro (the northern part of this area), local environmental conditions, such as an upwelling phenomena and large estuarine bays, together with contributions of continental drainage and anthropogenic activities, could determine different croaker populations. The aim of this study was to assess the fine-scale population structure of this species in three localities in Rio de Janeiro State (Brazil) and to compare it with previous studies that used different approaches. Through the combined use of genetic markers (nuclear microsatellites) and otolith signatures (morphometry and chemistry), two genotypic (North + Center/South) and three phenotypic (North + Center + South) populations were found. These results could contribute to a better understanding of the *M. furnieri* population dynamics and allow a rational management of this important fishing resource.

**Abstract:**

The inshore area of the Southwestern Atlantic between 22 °S and 29 °S (South Brazilian Bight) is a transitional climatic zone, where the tropical and warm temperate provinces mix. In its northern part, i.e., in the coastal waters of Rio de Janeiro, Brazil, local oceanographic conditions, such as upwelling in the north, and great bays with different degrees of anthropogenic influences in the center and south can determine the population structure of several fish stocks. The Whitemouth croaker (*Micropogonias furnieri*) is one the most heavily exploited fishing resources in this area, but there are still some doubts about its population structure. In this study, through combined analyses using nuclear genetic markers and morphological and geochemical signatures of otoliths, a divergence of individuals between two populations was identified using microsatellites, while a finer spatial structure with three populations (north, center and south, respectively) was found based on otolith shapes and elemental signatures. This regional population structure may have direct implications for rational fisheries management and conservation of the species.

## 1. Introduction

The whitemouth croaker *Micropogonias furnieri*, one of the most important fishery resources in Brazil, has been overfished since 1960 [1]. Although the species distribution area covers the entire Brazilian coast, the largest catches occur in the south-southeastern waters [1,2,3]. In the last few years, several studies using different methodologies have attempted to unravel the population structure of *M. furnieri* along the South American coast. Nevertheless, the results have been somewhat contradictory [1,4,5,6,7].

Regarding the main fishing area, the existence of one, two, or even three distinct population units of *M. furnieri* has been previously suggested [4,5,6,7,8,9,10,11,12]. A single south-southeasterner coastal population unit has been proposed using biochemical muscle analyses [8], alloenzyme electrophoresis [9] and mitochondrial DNA restriction patterns [6,10] (yellow line, Figure 1). However, two population units were proposed using data from the fisheries and reproductive biology: one distributed from Cabo Frio (23 °S) to Cabo de Santa Marta (29 °S); and another located between Cabo de Santa Marta and Chuí (33 °S), with close agreement between their boundary area [4,5,11] (green line, Figure 1). Later, another study using mitochondrial and nuclear genetic markers suggested the existence of a population unit between 23 °S (center of Rio de Janeiro State) to 29 °S (southern of Santa Catarina) and another one south of 29 °S [7], which closely coincides with the limits of the two populations previously suggested. The proposal for the existence of three population units was based on morphometric and age-determination analyses [1]. The study suggested the existence of one population between Chuí–Rio Grande do Sul (33 °S) and Santa Marta–Santa Catarina (29 °S), a second population between Santa Marta–Santa Catarina and Cabo Frio–Rio de Janeiro (23 °S), and a third population north of Cabo Frio–Rio de Janeiro (blue line, Figure 1). Moreover, using intron size polymorphism analysis (EPIC-PCR), a previous study [12] suggested the existence of a population in Rio de Janeiro (23 °S), another between Santos–São Paulo (24 °S) and Torres–Rio Grande do Sul (29 °S) and a third one between Torres and Chuí–Rio Grande do Sul (33 °S) (brown line, Figure 1). From the studies mentioned above, Rio de Janeiro, and especially the Cabo Frio region, appears to be the northern limit of the southeastern population of *M. furnieri* in the three studies [6,7,12] and one also pointed out the existence of another population north of Cabo Frio, although they did not present any results to corroborate it [1]. Therefore, the identification of their population units and associated stock boundaries, essential information for conservation requirements and sustainable fisheries management purposes, remains debatable.

Rio de Janeiro is in a transitional climatic zone [13,14,15] and exhibits peculiar local influences, such as a seasonal phenomenon in the north caused by the subtropical convergence waters which bring cold water to the surface on the north coast. The Cabo Frio region is considered by some authors as the limit and a transitional area between the tropical and temperate (or Patagonian) provinces [16,17]. The region is characterized by the existence of cooler outcrops that occur mainly between September and April, when the prevailing winds from the northeast transport waters from the uppermost layers to areas far from the coast, favoring the transport of lower layers deep into the surface layers [18]. This upwelling phenomenon, which brings nutrient-rich cold waters from the bottom to the surface, increases primary productivity and biomass [19]. Although the complexity of the coastal currents in this area makes it challenging to understand them as oceanographic barriers, the Cabo Frio upwelling area could act as a barrier to the dispersal of some species [20,21] (Figure 1).

Among the modern techniques used to study the population structure of fish, nuclear genetic markers and otolith morphology and geochemistry signatures have been of growing interest to scientists. Microsatellites have been widely used in fishery genetics to identify contemporary population structures, revealing subtler patterns than conventional mitochondrial markers [22,23,24]. Moreover, otoliths can be used as complementary natural tags to the molecular markers for studying marine fish populations living in water bodies with spatial heterogeneity. Factors such as extended larval period, large population size, and the absence of physical barriers favor high gene flow that masks genetic spatial differentiation [25,26,27]. Otoliths are calcified structures present in the inner ear of teleost fish with balance and hearing functions. They are species-specific and metabolic inert structures that grow continuously through the deposition of calcium carbonate in concentric layers and incorporate trace elements from the environment [28,29]. Otolith shape and chemical composition appear to be related to exogenous factors such as water salinity and temperature, but they could also be influenced by endogenous factors such as feeding regimes and ontogenetic changes [30,31,32,33]. The intrinsic characteristics of otoliths allow them the preservation of a chronological signature of the environment and therefore help us to identify population units [34,35], fish movements [36,37], habitat connectivity [38,39], and natal areas [40], among others.

The objective of this study was to assess, for the first time, through nuclear markers (microsatellites) and otolith analyses (elemental and morphological signatures), the microscale population structure of *M. furnieri* on the Rio de Janeiro coastal area, Brazil.

## 2. Materials and Methods

### 2.1. Study Area and Fish Sampling

The Southern Brazilian Bight (SBB) is the region along the southwestern Atlantic coast between the latitudes of 22 °S (Cabo de São Tomé–Campos dos Goytacazes/Rio de Janeiro) and 29 °S (Cabo de Santa Marta–Laguna/Santa Catarina). It is a transitional region between the tropical and temperate provinces. There are several upwelling areas that increase productivity across this area; one in the north of Cabo Frio (22.7 °S); another in the south near Cabo de Santa Marta (28.6 °S), and a third, less conspicuous, is near the Ilha Bela coast (23.8 °S). In these areas, seasonal upwelling takes place mainly between September and April, resulting in cold water temperatures that reach 18 °C [41].

Rio de Janeiro is situated in the north of the SBB and has different influences along its 650 km coast extension, encompassing different coastal systems determined by geomorphological and ecological characteristics. In the present study, three localities along the Rio de Janeiro coast, separated by approximately 400 km, were studied: (1) Macaé, located in the north of Cabo Frio (22.7 °S), is the region most influenced by the up-welling; (2) Itaipú (23 °S), located in the center, has the influence of Guanabara Bay, an anthropized estuarine system; and (3) Ilha Grande (23.2 °S), located in the south, has the most environmentally preserved system (Figure 1).

Priviously described hjhjhFor the microsatellite analyses, one hundred individuals of *M. furnieri* from an unpublished work [42] and historical collection were used. Adult individuals (unknown total length, TL) from the North (*n* = 18), Center (*n* = 45) and South (*n* = 37) were collected between August 2002 and May 2006 (for more details see Table 3.1 of [42]). A muscle portion of the fish dorsal region was removed and stored in ethanol 90%.

For the otolith analyses, ninety individuals were collected between September and October 2016: 30 from the North, 30 from the Center and 30 from the South. All sampled fish were adults (TL > size at first maturity) and ranged between 29.5 and 39.5 cm of TL (to 0.1 cm). Sagittal otoliths were carefully removed, measured (otolith length: OL, to 0.1 cm) and weighed (otolith weight: OW, to 0.001 g). Left otoliths were photographed for shape analyses and right otoliths were preserved clean and dry for elemental chemical analyses.

For both molecular and otolith analyses, fish were collected from the artisanal fishermen over several consecutive days until the desired sampling size was achieved, ensuring that the fishing area corresponded to the landing location.

### 2.2. Microsatellite Analysis

DNA from muscle tissue samples was extracted using the saline method [43]. Microsatellite genotyping was performed using the tailed primer method [44,45], and twelve microsatellite markers previously developed for *M. furnieri* were used [42]. Amplification reactions contained 200 M of each dNTP, 2.5 mM of MgCl_2_, 50 μg acetylated bovine serum albumin (BSA), 0.5 mM of each primer and 0.5 mM of the fluorescent tail, 1 unit of Taq polymerase (Sinapse Inc., Hollywood, FL, USA), 1X PCR buffer, and 30 ng of template DNA in a final volume of 15 μL per reaction. PCR thermocycling conditions were one cycle of 94 °C for 5 min followed by 30 cycles of 94 °C for 45 s, 60 °C for 45 s, 72 °C for 45 s. After these first amplification steps, eight more cycles at 94 °C for 45 s, 53 °C for 45 s and 72 °C for 45 s were performed to incorporate fluorescent-labeled primers (tail) into the amplified products. Genotyping, using GeneScan Liz 500 (Thermo fisher Sci. Inc, Waltham, MA, USA) as size standard, was performed in an automated sequencer ABI 3130 at Plataforma Genômica Multifuncional e Multiusuária (IBRAG/UERJ). It is noteworthy to highlight that samples from the North were genotyped and compared to genotypic data generated under the same conditions from a previous study [42].

To evaluate the occurrence of problems inherent to genotyping, such as the presence of null alleles and drop-out alleles, the Micro-Checker 2.2 (Hull, England) [46] was used. After correcting the genotyping problems, the raw data was converted to the input files of all programs used for analysis using Create 1.3.7 (Amherst, MA, USA) [47]. For linkage disequilibrium analyses between all loci pairs, the online version of the GENEPOP program (Montpellier, France) [48] (http://genepop.curtin.edu.au/ (accessed on 4 October 2021)) was used, with 10,000 iterations. A linkage disequilibrium analysis was also performed in Arlequin 3.5 (Bern Switzerland), which sought to identify the presence of associated alleles at a higher frequency than would be expected at random. To calculate the number of alleles of each loci, expected and observed heterozygosities, deviations from the expected Hardy-Weinberg equilibrium and inbreeding coefficient estimates (F_IS_) in the three populations, GeneAlex 6.5 (Acton, Australia) was used [49,50]. The Arlequin 3.5 [51] was also used to test population structures and compare gene frequencies through pairwise F_ST_ and Analysis of Molecular Variance (AMOVA). Genetic structuring was also evaluated with a Bayesian analysis using the STRUCTURE software (Chicago, IL, USA) [52]. This analysis seeks to estimate which number of K populations best explains the data obtained. The algorithm assigns individuals to populations, seeking to minimize Hardy-Weinberg imbalances within these populations. The STRUCTURE program was used for population data analysis. Ten independent runs were made for each of the K values. The K values analyzed ranged from 1 to 6 for the population data set. In each run, 100,000 burn-in replicas were used followed by 1,000,000 Markov chain replicas (MCMC) and was considered an admixture model. Another approach used to identify the most likely number of clusters (or populations) was the ΔK [53]. The ΔK statistic is defined as a second-order measure of the rate of change in the K-dependent probability function, and the highest ΔK value is expected to indicate the most likely K value. The ΔK values were calculated by the STRUCTURE Harvester program [54] from the data obtained in the STRUCTURE runs. The data obtained from the 10 independent STRUCTURE runs for the most likely K were compiled by the CLUMPP program [55] using the Greedy algorithm option, and the structure obtained was visualized in the DISTRUCT program (Stanford, CA, USA) [56]. A correspondence factor analysis (CFA) was used to visualize the distribution of genetic variation among individuals in a multidimensional space. This analysis was performed using Genetix 4.02 (Montpellier, France) [57]. Maximum likelihood estimates of gene flow between pairs of populations were performed with Migrate-n 5.0.4 [58,59]. This analysis was performed considering a full migration matrix model along 10 long chains with a burn-in of 1 × 10^5^ with 5 independent replicas. To identify the presence of relatedness between individuals (Unrelated, Half Siblings, Full Siblings, and Parent) within each location group, ML-RELATE software (Bozeman, MT, USA) was used [60]. This software uses maximum likelihood estimates to determine which relationships are consistent with genotype data and to compare potential relationships with alternative ones. The related pairs identified by the analysis were then subjected to a hypothesis test using a Likelihood Ratio test for two a priori relationships with a significance threshold of *p* < 0.05 [60].

### 2.3. Otolith Analysis

For the shape analyses, the otoliths were photographed on a black background with a Leica 205M stereomicroscope with a MC 170 HD camera and the software Leica Application Suite (LAS V2.0) (Leica Micro-systems, Wetzlar, Germany). Multivariate analysis of otolith shape was performed using elliptical Fourier descriptors (EFDs) according to a shapeR package routine [61]. The sequence of the package includes transforming the images into grayscale and binarizing them using a threshold pixel (a value of 0.3 was used in the Whitemouth croaker otoliths) to collect otolith outlines. The mean otolith shape of the three areas was also obtained. Basically, the analysis describes, in two dimensions, the otolith contours using the sum of sine and cosine functions and adjusting harmonics in relation to the real otolith contour [61]. Each individual consists of four harmonic coefficients (A, B, C and D) and the first 12 harmonics had enough power to represent the otolith outlines. The first three Fourier Coefficients (FCs: A1, B1 and C1) are constants resulting in 45 FCs to test the contours of the catch sites. These coefficients were logarithmized to attend to the parametric assumption and used to compare the three catch locations (PERMANOVA) and the possibility to re-classify the individuals according to the contours with the linear discriminant function analysis (LDFA).

For the chemical analyses, the otoliths were decontaminated with 3% ultrapure hydrogen peroxide (H_2_O_2_, Fluka TraceSelect) for 15 min, followed by 1% ultrapure nitric acid (HNO_3_, Fluka TraceSelect) for 10 s, and then double-washed with ultrapure water (H_2_0, Millli-Q -Water) for 5 min, and then dried in a laminar flow chamber [62]. Otoliths were thereafter weighed (OW, to 0.0001 g) and dissolved overnight with 10% ultrapure HNO_3_ to a final volume of 15 mL. The whole otolith samples were analyzed by inductively coupled plasma atomic emission spectrometry (ICP-OES) equipped with a cyclonic spray chamber and a Burgener MiraMist nebulizer, under the following conditions: forward power 1000 W, argon flow plasma 12 L/min, sheath gas 0.8 L/min. Otoliths were randomly analyzed to avoid sequence effects. The calibration curve was constructed using six standards made by successive dilutions with the Inorganic Venture multi-element LCA pattern, Christiansburg, VA, USA. The elements to be analyzed were pre-selected based on a previous otolith microchemistry study on this species [63]. Six elements (^42^Ca, ^88^Sr, ^137^Ba, ^55^Mn, ^24^Mg, and ^63^Cu) were consistently detected above the limit of detection limit. A certified reference material (FEBS-1) was used for quality analytical control [64]. The elemental concentrations determined in FEBS-1 were within certified values, and a recovery rate of 90% to 110% was obtained. The accuracy of the replicated analyzes for the individual elements was between 2% and 5% of the relative standard deviation (RSD). The detection limits of the calibration curves using the three-sigma criteria were (in ppb): ^42^Ca (100), ^88^Sr (4), ^137^Ba (20), ^55^Mn (20), ^24^Mg (20) and ^63^Cu (20). Geochemical signatures of otoliths were tested for normality (Shapiro–Wilk, *p* > 0.05) and homogeneity of variances (Levene, *p*> 0.05), and data Log(x + 1) transformed if necessary (Ba:Ca and Mn:Ca). To prevent any size/age differences in individuals from influencing the microchemistry results among locations, thereby confounding the spatial variation, an analysis of covariance (ANCOVA) was performed [38,65,66]. Thus, the relationships between the elemental ratios (Element:Ca) and the OW of otoliths were tested by ANCOVA. OW was considered a proxy for age and growth variation and was therefore used as a covariate in ANCOVA. The location was treated as a fixed factor. A One-way Analysis of Variance (ANOVA) was performed to test each Element:Ca ratio separately to identify differences among locations, followed by a Tukey post-hoc test, if needed (*p* < 0,05). Multivariate analysis of variance (MANOVA) and linear discriminant function analysis (LDFA) were used to explore the variation of multi-elemental signatures among locations [67]. For MANOVA, the Pillai statistic test was used. The reclassification accuracy of each site was assessed by the percentage of cases correctly re-classified through a Jackknife cross-validation analysis (leave-one-out). For a joint approach using multi-elemental signatures and EFDs, a PERMANOVA and a LDFA was performed using the package CARET in R. Statistics were calculated using Systat software 12 (Chicago, IL, USA) and R 4.1.3. Results are presented as mean ± standard errors. A significance level of *p* < 0.05 was adopted.

## 3. Results

### 3.1. Microsatellites

Some deviations from Hardy-Weinberg equilibrium in Mfur 02 (South), Mfur04 (North and South), Mfur07 (South), Mfur10 (South), Mfur12 (North and South), Mfur17(North), Mfur25 (South) and Mfur26 (South) were detected. However, none of the linkage disequilibrium analyses showed significant values, demonstrating independence between the data.

The suspicion of null alleles in markers Mfur02 in the Southern population, Mfur10 in the Center population, and Mfur12 in the North population (*p* < 0.05) was detected. The number of alleles per loci ranged from 6 at the Mfur20 loci to 33 at the Mfur03 loci with a mean value of 11.722 ± 1.105. Summary statistics for the microsatellite dataset is provided in Table 1.

All Pairwise F_ST_ comparisons were significant (*p* < 0.05), with higher values in comparisons between the north and the other locations (Table 2). Bayesian analysis performed in STRUCTURE suggests K = 2 as the more likely number of partitions in the dataset. The Q coefficient bar plot suggested that the North population is the most divergent from the others (Figure 2). This scenario of two populations is also supported by Correspondence factor analysis, which indicates similarity between individuals from the Central and Southern populations in contrast to those from the North (Figure 3). AMOVA analysis also supports this genetic structure scenario, with 10.8% of variance observed among groups (Φ_CT_ = 0.10828; *p* < 0.01), 0.21% among populations within groups, and 88.9% within populations.

Gene flow analysis showed an asymmetric migration pattern between pairs of locations (Table 3). Theta estimates were 7.20109 (CI 95%: 5.46667–11.46667) for the North, 2.94961 (CI 95%: 0.6667–5.0667) for the Center and 4.57987 (CI 95%: 1.33333–7.46667) for the South.

The mean relatedness (k) in the North was 0.035. Only one parent/offspring relationship, one full sibling relationship, and eight half-sibling relationships were confirmed by the hypothesis test (Likelihood Ratio test for two a priori relationships, *p* < 0.05). In the Center, the mean k was 0.042, with one parent/offspring relationship, 13 full sibling relationships, and 12 half-sibling relationships confirmed by the hypothesis test (Likelihood Ratio test for two a priori relationships, *p* < 0.05). In the South, k was 0.031, with all 13 half-sibling relationships significant (Likelihood Ratio test for two a priori relationships, *p* < 0.05). All other pairs of relationships were considered unrelated (total pairs comparisons: 134, 664, 990 for the North, Center, and South, respectively).

### 3.2. Otoliths

Otolith shape showed differences among the three locations (PERMANOVA, Pseudo-F = 5.08, df = 2, *p* < 0.01). These differences are reflected in the mean shape at different points as the rostrum and excisura (Figure 4). Overall, a total of 97% of individuals were correctly re-classified to the original site by the Jackknife classification matrix, ranging from 90% in the Center to 100% in the North and South (Table 4, Figure 4).

Sr:Ca and Mg:Ca were significantly different between the extreme locations (Figure 5), with Sr:Ca being higher in the South (One-way ANOVA: F = 23.212; df = 2.87; *p* < 0.001. Tukey-test: *p* < 0.001) and Mg:Ca in the North (One-way ANOVA: F = 34.336; df = 2.87; *p* < 0.001. Tukey-test: *p* < 0.001). The Ba:Ca (One-way ANOVA: F = 87.308; df = 2.87; *p* < 0.01. Tukey-test: *p* < 0.01), Mn:Ca (One-way ANOVA F = 95.665, df = 2.87; *p* < 0.001. Tukey-test: *p* < 0.001) and Cu:Ca (One-way ANOVA F = 155.724, df = 2.87; *p* < 0.001. Tukey-test: *p* < 0.001) ratios were higher in the Center compared with the other two locations (Figure 5). The multi-element signatures also showed differences among locations (MANOVA: Pillai’s test; F = 1.431; df = 10.168; *p* < 0.001). LFDA depicted a clear separation of the individuals among the 3 locations. Overall, 99% of the individuals were correctly re-classified to the original location by the Jackknife classification matrix, ranging from 97% in the North to 100% in the Center and South (Table 4).

The combined analyses of otolith shape and elemental signatures showed differences among locations (PERMANOVA: Pseudo F = 5.32, df = 2, *p* < 0.01) and a re-classification success of 100% for the entire dataset was achieved (LDA). Despite the result being similar between EFDs and ER, the three groups seem to be better separated using EFDs + ER (Figure 6).

## 4. Discussion

Combined analyses of microsatellites and otolith morphology and elemental signatures have allowed us to unravel, for the first time, the fine-scale population structure of Whitemouth croaker in Rio de Janeiro State Coast, Brazil.

The microsatellite dataset supports the occurrence of two genetic clusters with consistent genetic difference between individuals of the North and the other two locations. This scenario is supported by multiple lines of evidence, namely Bayesian analysis, as implemented in STRUCTURE, AMOVA, the correspondence factor analysis and the gene flow analysis, which showed an asymmetric migration pattern between pairs of locations. Values of the fixation index F_ST_ also suggest the occurrence of heterogeneity among samples of *M. furnieri* collected along the coast of Rio de Janeiro, but with slight differences. Higher values of F_ST_ were observed in the comparisons between the North and the other two populations, although a lower but significant value was observed between the Central and Southern populations.

The shape and elemental otolith signatures suggested a similar scenario, but differentiated all the three sampling locations, mainly as a result of a latitudinal gradient for Sr:Ca and Mg:Ca and higher concentrations of Ba:Ca, Mn:Ca and Cu:Ca in the central area. These results suggest that the individuals have inhabited enough time in different water masses throughout their lives and/or present different life history traits. The well-known upwelling at the North, the highly anthropized Guanabara Bay at the Center, and the environmental well preserved Ilha Grande Bay at the South, could somewhat explain these results in the Rio de Janeiro coastal waters.

Our main hypothesis to explain the existence of a clear separation between the North and the other localities is that the limit of the oceanographic provinces (Cabo Frio) is somehow acting as a barrier for the species gene flow, causing a genetic differentiation between the populations at the Northern and Central/Southern areas. The differentiation between these populations in Rio de Janeiro may be due to the intersection of currents, where tropical waters of the Brazilian current meet cold waters of the sub-Antarctic current, giving rise to waters of subtropical convergence or Central Atlantic South Waters (ACAS) [68]. These water bodies flow northwards below the Brazilian Current and seasonally come to surface, giving rise to the upwelling phenomenon [19,41]. One alternative explanation for the observed differences could be a bias caused by relatedness among the samples. However, the relatedness analysis indicated that there were relatively few significant interactions, representing less than 8% of the relationship pairs from each location. It is important to highlight that analysis of gene flow indicates that this barrier seems to be permeable at some level, probably in some seasons of the year where the effects of the upwelling are less pronounced.

Little is known about the permeability effect of the barrier for species distributed across the region, but Cabo Frio in the north of Rio de Janeiro is considered a border of congeneric dividing currents and species populations [13,69]. However, there are some previous findings about the effects of this oceanographic barrier on other fish species, such as the case of the Sciaenidae species *Macrodon atricauda* in the southeastern region of Brazil, which differs from the *M. ancylodon* species through shape analysis [70,71], the Mugilidae *Mugil liza* with a differentiated population unit in Rio de Janeiro inferred from microsatellites [69,72], and, more recently, a new population unit of the Clupeidae *Sardinella brasiliensis* identified by shape and multi-elemental analyses was found in Rio de Janeiro [34]. Genetic divergences in the populations of the Sciaenidae *Atractoscion aequidens* and *Argyrosomus inodurus* were also recorded between the northern and southern Benguela regions, an area known to have a consistent seasonal upwelling mechanism [73,74,75]. The structure of marine populations is usually related to the flow of currents, coastal topography, circulation patterns and water temperatures, but larval retention mechanisms, adult migrations patterns and species life histories also play an important role [70,73,75,76].

Otoliths were efficient tools in ichthyology and can give us clues as to whether fish live in different environments, experience different feeding regimes or have different genetic backgrounds [77,78,79,80,81]. In our study area, the Center is directly influenced by the continental runoff of Guanabara Bay, while the North is influenced by the upwelling with low water temperatures throughout most of the year. These differences could explain the results from both otolith shape and chemical signatures; if the populations are isolated from each other through a barrier or do not move between the studied locations for other reasons, or even if they have inhabited enough time in different water masses throughout their lives to have distinct natural tags.

The data recorded in fish otoliths showed higher concentrations of Mg:Ca in the North and Sr:Ca in the South. Such elemental concentrations present in otoliths are generally imposed by the water surrounding the environment, which constantly responds to variations caused by hydrogeological processes, tidal regimes, precipitation patterns and upwelling phenomena [82]. Upwelling currents bring from the sea floor elemental concentrations previously preserved in the sediment under aphotic conditions. The lower concentration of Sr:Ca found in the North may be related to the temperature difference caused by the upwelling [82,83]. However, the expected increase in the otolith Ba:Ca content also resulting from the upwelling was not observed in the present study. Instead, the higher Ba:Ca found in the Center was probably related to the influence of Guanabara bay and its continental water runoff with low salinity, as found in many studies focused on estuarine waters [29,38,84]. At the same time, this anthropogenic water input could explain the higher relation of Cu:Ca, namely in places contaminated with toxic metals [39,40], and Mn:Ca, with higher concentrations in hypoxic environments, although it also varies with salinity and temperature [85,86].

Several studies showed that the incorporation of vestigial and trace elements in the aragonite matrix of otoliths could be also influenced by physiological variables such as feeding regime, growth and ontogenetic differences and sex-specific variations [87]. That is why the combination of trace elements and contour shape analyses in otoliths has been used to identify spatial segregation in marine fish populations [72,88,89]. In addition, these natural tags are usually related to various abiotic and biotic factors that often allow us to differentiate individuals who occupy different water bodies and/or habitats [90,91,92]. The chemical composition of the whole otoliths provides a tag that integrates the entire life of the fish, from embryonic stages to capture, and can serve as a tracer of a particular group. This raw approach is useful for characterizing chemical tags of groups of fish, and then subsequently tracking the movement or mixture of those groups over time. The chemical composition will remain distinct and identifiable for a period of time, even if the groups move or mix, thereby accreting new otolith material with new chemistry, provided this additional material is minimal [84].

The microsatellite dataset corroborated the feeling that individuals found in Rio de Janeiro coastal waters are not genetically homogenous. The big challenge now is to identify what kind of barrier separates these populations (northern from central/southern individuals). The role of the upwelling phenomena as an oceanographic geographical barrier is well accepted for eggs and larvae that are passively dispersed through water currents [20,21,76]. However, juvenile and adult fish that swim actively would not have great dispersal difficulties. Moreover, *M. furnieri* adults can perform regular migrations from estuarine areas to continental shelf waters and should not suffer from such limitations. However, the upwelling occurs in the same period of the year in which the spawning of the species takes place in the state of Rio de Janeiro. This species spawns during spring, summer and autumn, coinciding with the upwelling of Cabo Frio in northern area of Rio de Janeiro [93]. This could act as an obstacle to the dispersal of eggs and larvae that will find suitable conditions here for the early life stages [75,76]. Furthermore, these low temperatures could alter physiological factors such as gonadal development and spawning, as this process is usually related to water temperature in fish species [94,95].

Among the obstacles that could cause constraints to the gene flow in marine environments, upwelling cells, environmental waters transitions, and sea surface temperature gradients are known to be incomplete/permeable barriers, leading to the divergence of allopatric populations and species [74,96,97]. Thus, it must be assumed that the difference between the water bodies’ temperatures in the study region, as a result of the Cabo Frio upwelling, is functioning as an effective barrier for the displacement of *M. furnieri* individuals.

The existence of a population structure in such a small-scale geographic area like that presented here is surprisingly, but not unprecedented, in *M. furnieri*. Indeed, a previous study found moderate to high genetic divergence among three populations of Whitemouth croaker sampled nearly 200 km apart long the Uruguayan Coast [35]. This highlights that other factors, such as local adaptation to environmental conditions including salinity and temperature, or the use of different bays and/or estuaries, can be important factors for differentiation in *M. furnieri*. Probably the apparent incongruence among the two population markers used in the current study is likely related to the evolutionary timescale at which each marker varies. Natural tags such as genetic markers are usually more conservative at broader spatio-temporal frames, whereas otolith chemical and morphological markers are usually environmentally dependent phenotypic traits [92]. However, more studies, with larger sample sizes collected in a broad distribution area during a short temporal window and designed specifically to disentangle the role of such environmental variables, are needed to elucidate the complex patterns of the population structure of this important fish species.

## 5. Conclusions

Nuclear markers suggested the existence of two populations of *Micropogonias furnieri* in the coast of Rio de Janeiro, geographically separated by Cabo Frio. However, otolith analyses have been able to discriminate the individuals of the three sampling regions in Rio de Janeiro, suggesting three populations instead. This study showed the importance of a holistic approach using genotypic and phenotypic complementary tools to study fish populations. New studies must be carried out to understand the cause behind the differentiation of these populations in this region. To continue to unravel this complex fish population structure, subsequent research should focus on the effects of the upwelling of the Cabo Frio region, the surface currents, the proximity of tropical regions or other environmental factors in its population dynamics.

## Figures and Tables

**Figure 1 biology-12-00360-f001:**
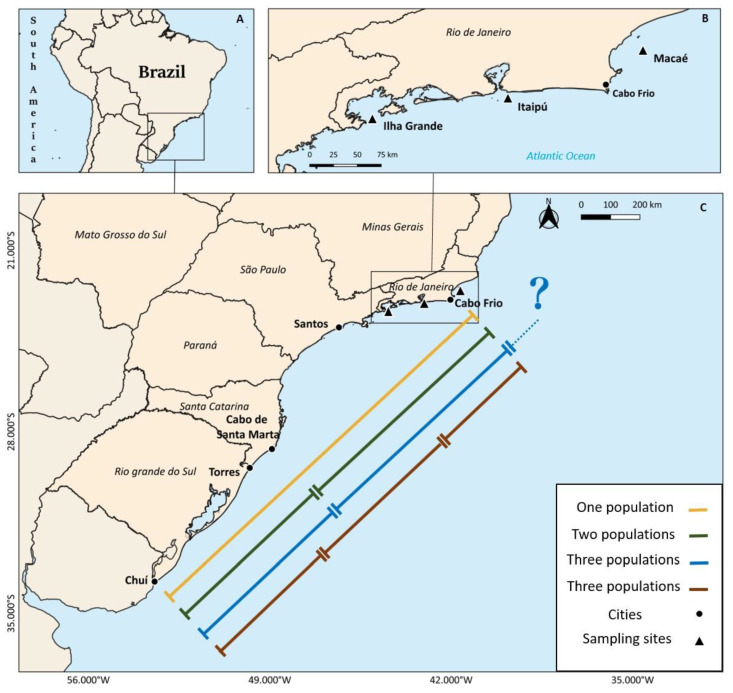
Location of Brazil in South America (**A**), and the southeastern‐south Brazilian coast (**B**), indicating the different population units of Micropogonias furnieri previously described (for further information, see Introduction). In the upper right corner (**C**), the map of the coast of the State of Rio de Janeiro, Brazil, is presented showing the three areas sampled in this study (black triangles): Macaé (North), Itaipú (Center) and Ilha Grande (South).

**Figure 2 biology-12-00360-f002:**
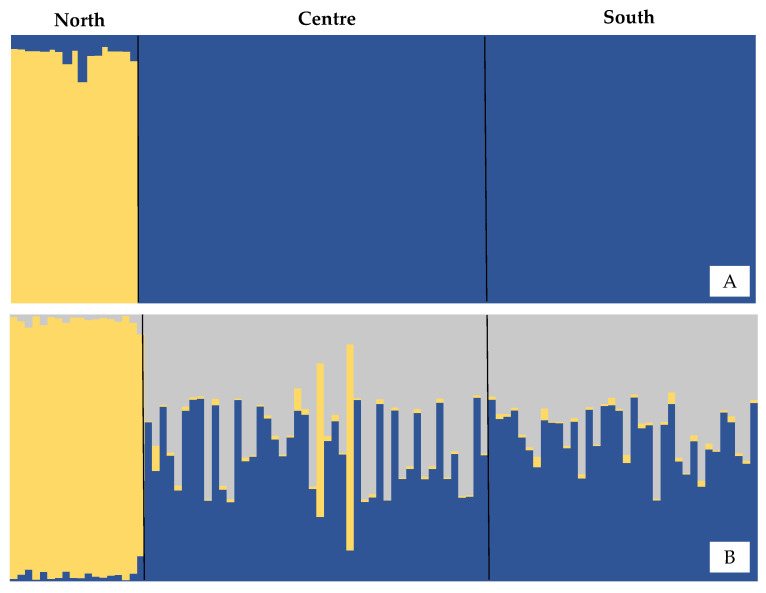
Q coefficient bar graph for each population, considering K = 2 (**A**) and K = 3 (**B**) population units for *Micropogonias furnieri*.

**Figure 3 biology-12-00360-f003:**
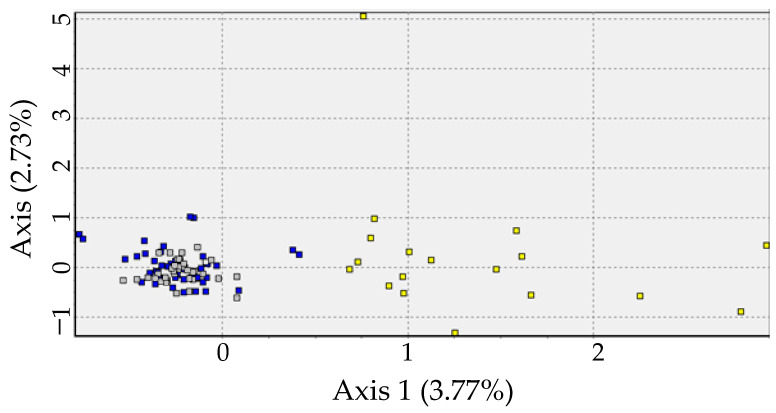
Relationship of the first two axes of the correspondence factor analysis per individual according to the microsatellite markers found in *Micropogonias furnieri* in Rio de Janeiro state. Yellow squares: North; grey squares: Center; blue squares: South.

**Figure 4 biology-12-00360-f004:**
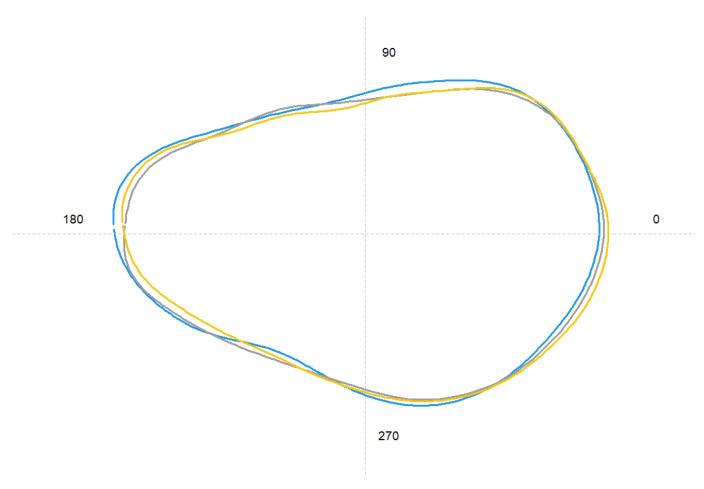
Average shape of otolith contour extracted from elliptical Fourier descriptors of the three sampling locations (yellow line: North; grey line: Center; blue line: South).

**Figure 5 biology-12-00360-f005:**
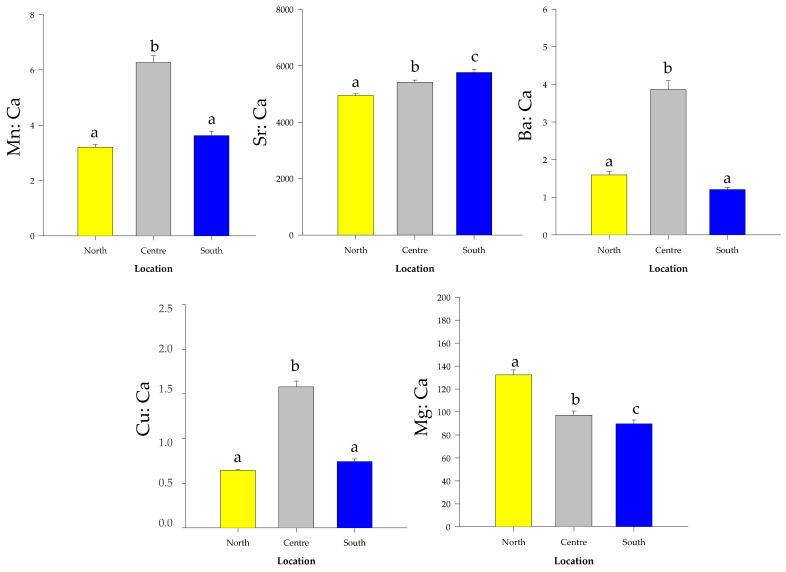
Element:Ca ratios (mean ± standard errors) of *Micropogonias furnieri* whole otoliths collected in the three sampling locations. The locations marked with the same letter above the error bars are not significantly different concerning the elemental concentrations (One-way ANOVA, followed by a Tukey test: *p* < 0.05).

**Figure 6 biology-12-00360-f006:**
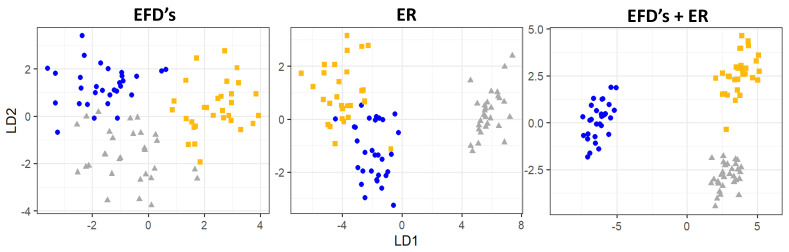
Scatterplot of the first and second discriminant function scores for the elliptical Fourier descriptors (EFDs), Element:Ca ratios (ER) and both (EFDs + ER) for *Micropogonias furnieri* individuals sampled in the study area (yellow squares: North; grey squares: Center; blue squares: South).

**Table 1 biology-12-00360-t001:** Basic statistics of microsatellite diversity in *M. furnieri* sampled at three sites at Rio de Janeiro State (Brazil). TNA = Total number of alleles at each loci; N = sampled individuals; NA = number of alleles; HO = Observed heterozygosity; HE = Expected Heterozygosity; HWE= Probability values of Hardy-Weinberg Equilibrium; FIS= Inbreeding coefficient estimates.

Locus	Mfur02	Mfur03	Mfur04	Mfur06	Mfur07	Mfur10	Mfur12	Mfur17	Mfur20	Mfur24	Mfur25	Mfur26
TNA	31	33	15	10	19	28	9	10	6	18	26	18
**North**												
N	12	10	12	11	16	5	11	12	11	12	15	9
NA	9	14	9	4	9	8	3	5	5	10	12	8
HO	0.583	0.700	0.500	0.545	0.500	0.800	0.273	0.250	0.727	0.667	0.800	0.444
HE	0.809	0.910	0.854	0.566	0.727	0.860	0.483	0.569	0.587	0.875	0.887	0.747
HWE	0.170	0.097	**0.013**	0.934	0.976	0.363	**0.010**	**0.003**	0.795	0.631	0.051	0.107
FIS	0.319	0.280	0.450	0.084	0.341	0.179	0.474	0.590	−0.194	0.279	0.132	0.453
**Center**												
N	35	34	33	37	36	37	35	31	36	32	27	34
NA	21	25	8	8	9	22	6	7	4	17	17	17
HO	0.857	0.971	0.727	0.432	0.722	0.946	0.657	0.581	0.361	0.844	0.852	0.824
HE	0.920	0.939	0.770	0.469	0.711	0.925	0.689	0.539	0.472	0.881	0.878	0.872
HWE	0.485	0.185	0.957	0.998	0.999	0.247	0.775	0.997	0.386	0.983	0.054	0.380
FIS	0.082	−0.019	0.071	0.091	−0.002	−0.008	0.060	−0.061	0.249	0.058	0.049	0.070
**South**												
N	40	41	44	43	39	40	32	45	45	35	44	42
NA	24	26	11	8	10	22	6	7	4	13	18	16
HO	0.875	0.902	0.705	0.395	0.795	0.800	0.750	0.600	0.422	0.829	0.932	0.929
HE	0.904	0.936	0.755	0.384	0.733	0.922	0.723	0.572	0.499	0.855	0.890	0.898
HWE	**0.000**	0.069	**0.000**	0.999	**0.000**	**0.000**	**0.003**	0.411	0.683	0.122	**0.000**	**0.000**
FIS	0.044	0.048	0.078	−0.017	−0.071	0.144	−0.022	−0.037	0.165	0.045	−0.036	−0.022

The **bold** values indicate significant differences (*p* < 0.05).

**Table 2 biology-12-00360-t002:** Pairwise FST values between the *Micropogonias furnieri* populations of the three different locations for the 12 microsatellite loci.

Location	North	Center	South
North	0.000	-	-
Center	**0.092**	0.000	-
South	**0.101**	**0.012**	0.000

The **bold** values indicate significant differences (*p* < 0.05).

**Table 3 biology-12-00360-t003:** Asymmetric estimations of gene flow between pairs of populations as calculated in Migrate. M: Mean value of the estimator of migrants per generation; CI 95%: 95% confidence intervals of estimates.

	To North		To Center		To South	
	M	(CI 95%)	M	(CI 95%)	M	(CI 95%)
**From North**	**-**	**-**	0.867	(0.573–1160)	1.081	(0.480–1233)
**From Center**	0.667	(0.393–0.933)	**-**	**-**	0.602	(0.573–1160)
**From South**	0.834	(0.480–1233)	0.789	(0.367–1.360)	**-**	**-**

**Table 4 biology-12-00360-t004:** Jackknife re-classification matrix based on Element:Ca ratios (ER), elliptical Fourier descriptors (EFDs) and both methodologies (ER + EFDs) for the North, Center and South locations.

ER		Predicted Site		% Overall Reallocation
Original Site	North	Center	South	
North	**100%**			97%
Center		**90%**	10%	
South			**100%**	
**EFDs**				
Original Site	North	Center	South	
North	**100%**			
Center		**90%**	10%	97%
South			**100%**	
**ER + EFDs**				
Original Site	North	Center	South	
North	**100%**			
Center		**100%**		100%
South			**100%**	

**Bold** values represent the correct reclassification success (%) for each site.

## Data Availability

Not applicable.

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
