# Peer review of "Identifying Whitemouth Croaker (Micropogonias furnieri) Populations along the Rio de Janeiro Coast, Brazil, through Microsatellite and Otolith Analyses"

_biology, 2023, doi:10.3390/biology12030360_

Round 1
Reviewer 1 Report
In this manuscript, Franco et al. assessed the population structure of M. furnieri along the Rio de Janeiro coast by the combined genetic markersand otolith signatures. Overall, this is a well-conducted study that adds several new insights into the M. furnieri population dynamics. However, I have several concerns as listed below. I pointed out some in Minor Comments below, but there are more errors to be corrected.
Materials and Methods
1.Why not use the same sample for molecular marker analysis and otolith analysis? In this case, the two methods do not corroborate each other. Moreover, 18 individuals from the North seems to be a bit less.
2. In the Microsatellites analysis section, it is necessary to add a description of the genotyping method. What about the 12 microsatellites polymorphism levels? Is such a small number of markers sufficient to accurately assess the level of genetic diversity of a population? Genotyping raw pictures or data are required in attachment.
Results
3. Genetic diversity indices of 12 microsatellite DNA markers for the three populations of M. furnieri should be support as a TABLE, including alele number, efective alele number, observed Heterozygosity, expected Heterozygosity, polymorphism index content, and P- value for Hardy-Weinberg test.
4. Fst is an important indicator to measure genetic differentiation between populations. The results of inbreeding coefficient (Fit), intrapopulation inbreeding coefficient (Fis), interpopulation differentiation coefficient (Fst) should be support in Tbale1. What proportion of genetic variation comes from between populations?
Discussion
5. The results of molecular markers are partially insufficiently discussed and do not compare with published results, more like conclusions.
Author Response
Reviewer 1#
1. Why not use the same sample for molecular marker analysis and otolith analysis? In this case, the two methods do not corroborate each other. Moreover, 18 individuals from the North seems to be a bit less.
R: Suggestion accepted. We used indeed existent historical collections of muscular tissue for genetics, and later on contemporaneous new samplings for otoliths in the same locations. Anyway, both methodological approaches (genotypic vs phenotypic) are not directly comparable, because they depend of different time-frame scales. Moreover, the sample size of the northern location is a smaller sample than that carried out in the other two regions for genetics purposes. However, we believe that this sample size is sufficient to represent the genetic diversity of the sampled are, once it is a similar sample size for some localities in other previous genetic studies with microsatellites (e.g., Mai et al. 2014, Vasconcellos et al. 2015). However, this situation is now clarified in M&M (See Revised MS, L142-146 and L154-155) and shortly discussed (See Revised MS L436-450). A new supporting reference is included (See Revised MS, L818-821).
2. In the Microsatellites analysis section, it is necessary to add a description of the genotyping method. What about the 12 microsatellites polymorphism levels? Is such a small number of markers sufficient to accurately assess the level of genetic diversity of a population? Genotyping raw pictures or data are required in attachment.
R: Suggestion accepted. Information regarding the amplification and genotyping was inserted in the revised MS (See Revised MS, L157-170). Polymorphism levels for microsatellite loci were included in a new table (See Revised MS, Table 2). Anyway, in general, 12 markers are a reasonable number of markers, allowing the analysis of genetic structure of populations. It is not uncommon the use of even less markers with successful detection of population structure in commercial fish species (e.g. Mai et al. 2014 – reference already cited in the original MS), including for M. furnieri (e.g., D`Anatro et al. 2011 – new reference included in the revised MS, L813-817).
3. Genetic diversity indices of 12 microsatellite DNA markers for the three populations of M. furnieri should be support as a TABLE, including allele number, effective allele number, observed Heterozygosity, expected Heterozygosity, polymorphism index content, and P- value for Hardy-Weinberg test.
R: Suggestion accepted. A new table was included. See Revised MS, table 2.
4. Fst is an important indicator to measure genetic differentiation between populations. The results of inbreeding coefficient (Fit), intrapopulation inbreeding coefficient (Fis), interpopulation differentiation coefficient (Fst) should be support in Table1. What
proportion of genetic variation comes from between populations?
R: Suggestion accepted. Information about pairwise FST comparisons was already in Table 1. But FIS estimates are now presented in new table 2. See Revised MS, Table 2.
5. The results of molecular markers are partially insufficiently discussed and do not compare with published results, more like conclusions.
R: Suggestion accepted. We improved the discussion about population structure of M. furnieri found here in the light of the literature and molecular markers. See Revised MS, L338-346 and 436-450.
Reviewer 2 Report
The manuscript claims population subdivision of the whitemouth croaker based on genetic (microsatellites), morphometric and biochemical evidence. The manuscript also attributes the subdivision to the regional up-welling. However, I think the results presented in the manuscript are not sufficient to support the conclusions that the authors made.
First, the study has a very small sampling area (three locations ~400km apart) comparing with the distribution of the whitemouth croaker (from Costa Rica to Argentina). It is hard to imagine a wide-spread fish species of median size can have such fine population structure. A previous study (Vasconcellos et al., 2015), with larger sampling area (covering almost entire coastal Brazil), indicated clinal differentiation, which I think it is a more creditable population structure.
Second, with the results, authors do not clarify criteria of delimiting populations. For example, pairwise FST of 0.092 to 0.101 between North and Centre/South, is not a sufficient value in the population genetics to delimit populations. Moreover, according to the PCA analyses (Figure 3), the genetic differentiation within North seems greater than the ones between some of the North individuals and the Centre/South ones.
Third, authors should provide more details in the sampling methods and basic statistics (heterozygosity, allelic richness etc.) of the microsatellite data. Otherwise, it is difficult to rule out the possibility that the differentiation observed is an artifact due to the high relatedness among individuals within each sampling location.
Therefore, I would recommend not to accept this manuscript unless authors can provide sufficient evidence to support their claims. For example, including samples outside the region of up-welling to identify if there is the division situated at the up-welling.
Author Response
Reviewer 2#
1. First, the study has a very small sampling area (three locations ~400km apart) comparing with the distribution of the whitemouth croaker (from Costa Rica to Argentina). It is hard to imagine a wide-spread fish species of median size can have such fine population structure. A previous study (Vasconcellos et al., 2015), with larger sampling area (covering almost entire coastal Brazil), indicated clinal differentiation, which I think it is a more creditable population structure.
R: Suggestion accepted. Our analyses based on both genetic and otolith data, although with slight differences, suggest the existence of this fine-scale population structure in the study area. Moreover, we are dealing with different but complimentary approaches in terms of spatial-temporal discrimination power. For marine fish species, genotypic methods are sometimes relatively weak compared to the phenotypic ones to find spatial discrimination (this was already highlighted in the original version, See Revised MS L100-113). Anyway, to clarify this issue, results are now much detailed, discussion improved pointing to the need of further studies. See Revised MS (look all red color changes).
2. Second, with the results, authors do not clarify criteria of delimiting populations. For example, pairwise FST of 0.092 to 0.101 between North and Centre/South, is not a sufficient value in the population genetics to delimit populations. Moreover, according to the PCA analyses (Figure 3), the genetic differentiation within North seems greater than the ones between some of the North individuals and the Centre/South ones.
R: Suggestion accepted. We fully understand the referee concerns, but multiple lines of evidence brought to different methods, namely FST, AMOVA, Bayesian analysis and PCA, supports the separation of populations. Furthermore, otolith analysis also supports divergence among populations. Some studies with microsatellite markers in widespread species of fishes found even smaller values of FST in comparisons among populations (e.g., Mai et al. 2014, Vasconcellos et al. 2015). PCA analysis presented in Figure 3 in fact shows a higher level of divergence within North population. However, this finding does not invalidate our conclusions since there is no overlap between individuals from the north and other locations. Anyway we included more detailed results and shortly discuss it. See new table 2. See Revised MS, L338-346 and L436-450.
3. Third, authors should provide more details in the sampling methods and basic statistics (heterozygosity, allelic richness etc.) of the microsatellite data. Otherwise, it is difficult to rule out the possibility that the differentiation observed is an artifact due to the high relatedness among individuals within each sampling location.
R: Suggestion accepted. We included more details about sampling methods in “M&M” and more information about basic statistics in “Results”. A new table 2 was included. Furthermore, we inserted more information about amplification and genotyping of microsatellite also in “M&M”. See Revised MS, L142-146, L154-155, L157-170, L179-181, L259-261, L266-268, L275-278. See new Table2.
Round 2
Reviewer 2 Report
Thank you for the effort in revising and providing additional materials for the manuscript.
Even though it is much better than the previous version. I still have a few concerns in the claiming of two to three populations (fine population structure) in the manuscript.
First, I'm not an expert in the chemical part. But I assume that the chemical composition is highly environment-dependent. This means, if catch a fish at location A and keep at location B, after a while, the fish will have a chemical composition similar to all the fish at location B right? Please justify the use of chemical composition as a criteria in identifying populations.
Second, "fish were collected from the artisanal fishermen" didn't rule out the possibility that the samples were collected at a single session and contain significant proportion of closely related individuals. I'd suggest to calculate the kinship of samples to inspect if there can be bias from sampling closely related family members.
Third, authors claimed that the population differentiation could be results of environmental barriers. To support this, I'd suggest authors to estimate the geneflow among populations (using programs such as Migrate-n), which can be a more intuitive evidence to infer the cause of differentiations.
In addition, there are also minor comments which I have inserted in the attached pdf with highlight and edits.

Author Response
"Please see the attachment."

Round 3
Reviewer 2 Report
I'd like to thank author to response my comments in details. Overall, the changes made have addressed some of my concerns on the interpretations of data in the manuscript.
I'm not very supportive in claiming three distinct populations with such shallow genetic divergence and high volume of gene flow. However, the delimitation of populations are always subjective on other factors.
Moreover, I don't see much discussion on why authors attribute the population divergence to the upwelling influence zone. If the claim is not supported by any observations (for example, fish at upwelling zone are much larger in size so created this differentiation), I'd suggest do not highlight in the abstract (because many readers may pickup information from abstract as a relatively sound conclusion).
In addition, I have two minor issues:
1. Authors claimed that the fish were caught in consecutive days and "unlike" to be related. However, I don't regard it as a solution to my comment. I have proposed simple analyses to calculate pairwise kinship, which I think is a better way to rule out potential bias in the dataset.
2. Authors need to explain what are the numbers in the Table 3. Migration rate in terms of effective population size or in terms of percentage?
Author Response
"Please see the attachemnt"
